# Effects of Encapsulated Methionine on Skeletal Muscle Growth and Development and Subsequent Feedlot Performance and Carcass Characteristics in Beef Steers

**DOI:** 10.3390/ani11061627

**Published:** 2021-05-31

**Authors:** Jessica O. Baggerman, Alex J. Thompson, Michael A. Jennings, Jerilyn E. Hergenreder, Whitney Rounds, Zachary K. Smith, Bradley J. Johnson

**Affiliations:** 1Department of Animal and Food Sciences, Texas Tech University, Lubbock, TX 79430, USA; jbaggerman@huntington.edu (J.O.B.); AThompson@zinpro.com (A.J.T.); hiredhandsouthdowns@hotmail.com (M.A.J.); 2Kemin Industries, Des Moines, IA 50317, USA; Jerilyn.hergenreder@kemin.com (J.E.H.); whitney.rounds@kemin.com (W.R.); 3Department of Animal Science, South Dakota State University, Brookings, SD 57007, USA; zachary.smith@sdstate.edu

**Keywords:** amino acid, beef cattle, methionine, satellite cells

## Abstract

**Simple Summary:**

Research investigating the effects of amino acids in growing and finishing beef cattle is not a new topic. Many studies suggest that in cattle fed a concentrate-based diet, lysine is the most limiting amino acid followed by methionine. When considering supplementing amino acids to growing-finishing beef cattle, the amino acids must be protected from ruminal microbial consumption for the beef animal to be able to absorb the amino acids in the small intestine. The regulation of skeletal muscle growth is a multifaceted, complex process controlled through many signaling cascades. One major process involved in muscle growth is the mammalian target of the rapamycin (mTOR) signaling pathway. The objectives of these studies were to investigate the role of encapsulated methionine on feedlot growth performance and carcass characteristics and to elucidate the role of methionine supplementation on the mTOR signaling pathway and skeletal muscle development in feedlot steers during the finishing phase.

**Abstract:**

Two studies were conducted to evaluate the effect of encapsulated methionine on live performance, carcass characteristics, and skeletal muscle development in feedlot steers. In Experiment 1, 128 crossbred steers (body weight [BW] = 341 ± 36.7 kg) were used in a randomized complete block design and supplemented with 0, 4, 8, or 12 g/(head day [d]) of ruminally protected methionine (0MET, 4MET, 8MET, and 12MET, respectively) for 111 d or 139 d. In Exp. 2, 20 steers (BW = 457 ± 58 kg) were stratified by BW and randomly assigned to either the 0MET or 8MET treatment; longissimus muscle (LM) biopsies were collected on d 0, 14, 28, 42, and 56, and analyzed for mRNA and protein expression. Additionally, immunohistochemical analysis was performed to measure fiber type area and distribution as well as the density of muscle nuclei and satellite cells (Myf5, Pax7, and Myf5/Pax7). In Experiment 1, no significant differences were observed for live performance (*p* ≥ 0.09). There was, however, a linear relationship between LM area and methionine supplementation (*p* = 0.04), with a 9% increase in the area when steers were supplemented with 12MET compared to 0MET. In Exp. 2, There were no treatment × day interactions (*p* ≥ 0.10) for expression of mRNA or protein abundance. Although mRNA expression and protein abundance of all genes were influenced by day (*p* ≤ 0.04), methionine supplementation did not have a significant effect (*p* ≥ 0.08). There was a significant treatment × day interaction for distribution of MHC-I fibers (*p* = 0.03), where 8MET supplemented cattle had a greater proportion of MHC-I fibers after 56 d of supplementation than did 0MET steers. Cross-sectional area was increased over time regardless of fiber type (*p* < 0.01) but was unaffected by treatment (*p* ≥ 0.36). While nuclei density was not impacted by treatment (*p* = 0.55), the density of myonuclei increased nearly 55% in 8MET supplemented cattle (*p* = 0.05). The density of Myf5 positive satellite cells tended to decrease with methionine supplementation (*p* = 0.10), while the density of Pax7 expressing cells tended to increase (*p* = 0.09). These results indicate that encapsulated methionine supplementation may influence markers of skeletal muscle growth, and potential improvements in the LM area may exist.

## 1. Introduction

There has been previous research investigating the effects of amino acids in growing beef cattle [1]. These studies suggest that in cattle fed a concentrate-based diet, lysine is the most limiting amino acid followed by methionine [2,3]. When considering supplementing amino acids to growing beef cattle, the amino acids must be protected in some fashion, such as encapsulation, from ruminal microbial consumption for the beef animal to be able to absorb the amino acids and utilize them for potential growth. In addition to being the most limiting amino acids in cattle, lysine and methionine are crucial for muscle synthesis [4]. Hosford et al. [5] reported an increase in growth performance when feeding encapsulated lysine and methionine with a beta-agonist and indicated that amino acid requirements may not be fulfilled by the typical diet in finishing cattle.

The regulation of skeletal muscle growth is a multifaceted, complex process controlled through many signaling cascades. One major process involved in muscle growth is the mammalian target of the rapamycin (mTOR) signaling pathway. While it is understood that many factors play a role in the regulation of mTOR signaling, the role of amino acids on mTOR regulation is still unclear [6]. Previous research suggests that amino acids may regulate mTOR through Rag GTPases (RAG). Amino acids stimulate RAG, which then may cause downstream upregulation of mTOR which could lead to increased skeletal muscle growth. As skeletal muscle grows, the cross-sectional area of skeletal muscle fibers increases, there is a change in skeletal muscle fiber type distributions, and skeletal muscle nuclei and satellite cell populations fluctuate [5]. By stimulating factors in the mTOR signaling cascade, we hypothesized that increased transcription due to methionine supplementation may occur and lead to improved growth performance.

The objectives of these studies were to investigate the role of encapsulated methionine on feedlot performance, carcass characteristics, the mTOR signaling pathway, and skeletal muscle development in feedlot steers during the finishing phase.

## 2. Materials and Methods

### 2.1. Experiment 1

#### 2.1.1. Animals

Two hundred and sixty-four crossbred steers (body weight [BW] = 355 ± 44.9 kg) were received at the Texas Tech University Beef Center in February 2014. Upon arrival, cattle were placed in receiving pens (*n* = 12) and provided ad libitum access to water, grass hay, and a 65% concentrate ration. Upon trial initiation (day [d] 0) steers were individually weighed (Silencer chute; Moly Manufacturing, Lorraine, KS, USA; mounted on Avery Weigh-Tronix load cells, Fairmount, MN, USA; readability ± 0.45 kg). Initial processing included the following procedures: application of a unique identification tag, vaccinations against viral (Vista 3; Merck Animal Health, Summit, NJ, USA) and clostridial (Vision 7; Merck Animal Health, Summit, NJ, USA) diseases, treatment for internal parasites (Safeguard; Merck Animal Health), and application of a terminal implant (Revalor-XS; Merck Animal Health).

One hundred twenty-eight steers (initial BW = 341 ± 36.7 kg) were selected from the larger population based on uniformity of BW and blocked (*n* = 8) by day (d) 0 BW. Within each block, steers were randomly assigned to the pen (*n* = 4), and 1 of 4 treatments were assigned to each pen within a block. Steers were sorted into 32 concrete, partially slotted-floor pens (4 steers/pen; 2.9 × 5.5 m with 2.4 m of linear bunk space). Steers received step-up rations (65, 75, and 85% concentrate) over the first 3 weeks of the trial period. The finishing ration (Table 1) was introduced on d 21 and was fed to provide ad libitum access to feed once daily in the morning for the duration of the experiment.

#### 2.1.2. Treatments and Experimental Design

Four treatments were used in a randomized complete block design. Treatments consisted of 4 concentrations of supplemental methionine supplied to the small intestine daily which will subsequently be defined as follows: 0MET = 0 g/(head d), 4MET = 4 g/(head ·d), 8MET = 8 g/(head d), and 12MET = 12 g/(head d). Methionine supplemented cattle were administered 0.00, 12.24, 24.49, or 36.73 g of methionine supplement (MetiPEARL; Kemin Industries, Inc., Des Moines, IA, USA) which contained 0.00, 6.73, 13.47, or 20.20 g of DL-methionine for the 0MET, 4MET, 8MET, and 12MET treatments, respectively. Ruminal bypass of the methionine supplement was assumed to be 66% with 90% digestibility (W. Rounds, Kemin Industries, Inc., Des Moines, IA, USA, personal communication, November 6, 2013) to target absorption in the small intestine of 0, 4, 8, or 12 g/(head d) of supplemental methionine. In addition, all cattle were supplemented with 4 g/(head d) of supplemental lysine (LysiPEARL; Kemin Industries, Inc., Des Moines, IA, USA) to meet requirements for the most limiting amino acid, with the methionine supplementation levels selected to be proportional to lysine supplementation.

#### 2.1.3. Treatment Application and Routine Management

Diets were formulated to meet or exceed National Research Council, 1996 [7] requirements for growing-finishing beef cattle and were prepared in the Texas Tech Burnett Center Feed Mill. Feed bunks were evaluated at approximately 07:30 h daily to estimate orts and adjust feed calls to ensure ad libitum access to feed [7]. The feed bunk management approach was to achieve ≤ 0.45 kg of dry orts in the bunk each day. Diets were mixed in a paddle type mixer, transferred by drag chain conveyor to a tractor-pulled mixer (Rotomix, Dodge City, KS, USA), and delivered once daily at 0900 h. Immediately following feed delivery, each pen was top-dressed with 0.45 kg of the respective treatment premix beginning on d 1. All treatment premixes were made at the Texas Tech University Burnett Center Feed Mill and included ground corn and supplemental methionine. Experimental diets included Rumensin (30 g/ton DM; Elanco Animal Health, Greenfield, IN, USA and Tylan (10 g/ton DM; Elanco Animal Health), and all cattle received a beta-agonist (RH; Optaflexx; Elanco Animal Health; 300 mg/(head d)) for the final 28 days on feed.

Weekly diet samples were obtained directly from the feed bunk immediately following feed delivery and frozen for subsequent analysis. Following trial termination, samples were composited within the interim period and submitted for proximate analysis (Servi-Tech Laboratories, Amarillo, TX, USA). Nutrient analysis and amino acid profile can be found in Table 1 and Table 2, respectively. Separate weekly samples were also taken and dried in a forced-air oven at 100 °C for 24 h to determine DM content which was used to determine total dry matter intake (DMI).

Unshrunk BW measurements were taken prior to feed delivery (0700) at 28-d intervals, beginning on d 0. Individual BW measurements were taken on d 0, prior to RH initiation, and before shipment to slaughter. Other interim BW measurements were obtained on a certified pen scale (readability ± 2.3 kg) on d 28, 56, and 84. Before collection of BW measurements, feed bunks were cleaned of residual feed. Orts were weighed and sampled for DM content. The DMI of each pen was adjusted to reflect the total DM delivered to each pen after subtracting the quantity of dry orts. Three steers were removed from the trial during the conduct of the study.

#### 2.1.4. Carcass Evaluation

When study personnel deemed approximately 60% of the cattle within a BW block to have an external fat cover sufficient to grade USDA Choice, they were transported to a commercial abattoir (Tyson Fresh Meats, Amarillo, TX, USA). Shipping occurred on d 128 for the heaviest 4 BW blocks and on d 140 for the remaining 4 blocks. During the harvest process, extra carcass trims, and fat and hide pulls of soft tissue ≥ 6.8 kg, were noted. Individual carcass measurements (*n* = 125) included hot carcass weight (HCW), 12th rib fat depth, LM area, kidney/pelvic/heart fat (KPH) %, and marbling score. Yield grade was calculated using the USDA regression equation [8]. A 3% shrink was applied to the final live weights for calculation of dressing percentage.

#### 2.1.5. Fabrication and Shear Force Analyses

A total of 61 strip loins (Institutional Meat Purchasing Specification #180; [9]) were obtained from carcasses during the first harvest group and transported to Texas Tech University for subsequent shear force analysis. Two days post-harvest, 1 steak was cut from the anterior end of each strip loin to level the cut surface. Four 2.54-cm steaks were cut from the anterior end and assigned to 1 of 4 aging periods (7, 14, 21, and 28 d) in rotating order to ensure each aging period was equally represented among anatomical position within the loin. Fabricated steaks were vacuum packaged within the same aging period and stored at 2 °C for the assigned aging period. After the appropriate postmortem aging period, steaks were frozen at –20 °C until further analysis.

Steaks were thawed at 4 °C for 24 h prior to cooking. Steaks were cooked to an internal temperature of 71 °C using a belt grill (Magigrill TBG-60; Magi-Kitch’n Inc., Quakertown, PA, USA) with a grill-plate temperature of 163 °C. Individual steak temperature and weight were recorded immediately before and after cooking. Steaks were cooled overnight to 2 to 4 °C. Warner-Bratzler shear force (WBSF) values were obtained by removing six 1.3-cm cores from each steak parallel to the muscle fiber. Cores were sheared once, perpendicularly to the muscle fibers, using a WBSF analyzer (G-R Elec. Mfg., Manhattan, KS, USA). The WBSF values from the six cores from each steak were averaged for statistical analysis.

#### 2.1.6. Statistical Analysis

Data were analyzed using the GLIMMIX procedure of SAS (SAS version 9.4; SAS Inst., Cary, NC, USA). For all live performance and carcass characteristic analyses, the pen was considered the experimental unit. Treatment was included as a fixed effect. Weight block and harvest date were considered random effects. For WBSF analysis, strip loin was the experimental unit, treatment was included as a fixed effect, and the block was considered a random effect. Single-degree-of-freedom preplanned contrasts were used to compare (1) 0MET vs. all other treatments, and (2) linear and quadratic effects of methionine inclusion. For all analyses, an alpha level ≤ 0.05 was considered significant, and tendencies were declared for values between 0.05 and 0.10.

### 2.2. Experiment 2

#### 2.2.1. Animals

Steers used in Exp. 2 were selected from the larger group of cattle received and processed in Exp. 1. All initial processing followed procedures described previously; however, the animals that were not selected for use in Exp. 1 were not administered an implant. All remaining cattle were weighed 7 d prior to initiation of Exp. 2. Twenty steers (BW = 456 ± 8 kg) were selected from the larger population based on uniformity of BW and split into 2 weight blocks. Each block consisted of 2 pens, and the pen within a block was randomly assigned to 1 of 2 treatments. Upon initiation, steers were sorted into 4 dirt floor pens with a concrete feed apron (5 steers/pen; 8 × 18 m with 8 m of linear bunk space).

#### 2.2.2. Treatments, Experimental Design, and Routine Management

Two treatments were used in a randomized complete block design. Treatment factors consisted of 2 concentrations of supplemental methionine (0MET or 8MET) as previously described in Exp. 1. Steers were provided ad libitum access to feed once daily in the morning for the duration of the experiment. Diets used in Exp. 2, as well as routine daily management strategies and application of treatments, followed the same procedures outlined in Exp. 1. At approximately 0800 h on d 0, 14, 28, 42, and 56 of treatment administration, LM biopsy samples, and individual BW were collected from all animals.

#### 2.2.3. Live Muscle Biopsy

Muscle biopsy procedures have been described previously [10,11]. Animals were secured in a hydraulic squeeze chute where the site of incision (11th to 13th rib) was prepared by removing all hair and debris from the hide using electric clippers and disposable razors. The site was then (1) washed with water and povidone-iodine surgical scrub (Betadine, 7.5% povidone-iodine; Purdue Products, Stamford, CT, USA), (2) rinsed with 70% ethanol, (3) shaved with a disposable razor for a second time, (4) washed again with water and surgical scrub, and finally, (5) rinsed with 70% ethanol. A local anesthetic (lidocaine HCl; 20 mg/mL; 8 mL per biopsy) was injected into the subcutaneous fat layer, in a 6 cm^2^ diamond-shaped pattern (4 injection sites, 2 mL lidocaine HCl per site), and given a minimum of 5 min to numb the area. The site was then rinsed again with ethanol and wiped with sterile gauze before a 1 cm incision was made with a sterile scalpel. A sterile 4-mm diameter Bergstrom biopsy needle was used to collect approximately 2 g of longissimus tissue. The extracted tissue was placed on a sterile gauze pad in a plastic container with a protective lid. The incision was then sealed using VetBond (3M Animal Care Products, St. Paul, MN, USA), and sprayed with AluShield (Neogen Corp., Lexington, KY, USA) to prevent infection.

Each biopsy sample was divided into three fractions. One fraction, designated for gene expression analysis by reverse transcription-polymerase chain reaction (RT-qPCR), and another fraction for protein analysis by SDS-PAGE and Western blotting were placed in Whirl-Pak (Nasco, Fort Atkinson, WI, USA) bags and snap-frozen in liquid nitrogen. The third fraction was used for immunohistochemical staining. Samples were analyzed with a magnifying glass and muscle fibers were identified within the sample. Fibers were oriented parallel to each other on a 1 × 1.5 cm cork board and frozen in a clear section compound (VWR International, West Chester, PA, USA) using isopentane and dry ice. All samples were placed in Whirl-Pak bags (Nasco, Fort Atkinson, WI, USA) and stored in a cooler with dry ice for transportation to Texas Tech University, where all samples were stored at −80 °C until further analysis.

Throughout the study, the biopsy incision site was alternated between sides of the animal, beginning at the 11th to 12th rib interface and moving anterior to the next rib interface on consecutive biopsy dates. Steers were monitored for 72 h following each biopsy for inflammation or infection of the incision site.

#### 2.2.4. RNA Isolation and RT-QPCR

Ribonucleic acid was isolated from biopsy samples using ice-chilled TRI-reagent (Sigma, St. Louis, MO, USA). Approximately 0.5 g of tissue was homogenized in liquid N_2_, and following the evaporation of N_2_, 3 mL of TRI-reagent was added. The resulting liquid fraction was extracted, placed in two 1.5-mL centrifuge tubes, and combined with 200 μL of chloroform. Tubes were then vortexed, centrifuged at 15,000× *g* for 15 min, and the upper aqueous layer was transferred to a new microcentrifuge tube. Five hundred microliters of chilled isopropanol was added to each tube, which was subsequently vortexed and centrifuged at 15,000× *g* for an additional 10 min. The supernatant was discarded, 500 μL of 75% ethanol was added to each tube, and samples were stored at −80 °C until further processing. Thawed samples were centrifuged at 15,000× *g* for 10 min at room temperature, and the pellet was retained. The pellet was then dissolved in 20 µL of nuclease-free water and the concentration of RNA was determined by spectrophotometry at an absorbance of 260 nm (NanoDrop; NanoDrop Products, Wilmington, DE, USA). Samples were purified using an Ambion DNA-free kit (Life Technologies, Carlsbad, CA, USA), and 1-µg of total RNA was utilized for reverse transcription to form cDNA. Real-time quantitative polymerase chain reaction (GeneAmp 7900HT; Applied Biosystems, Grand Island, NY, USA) was performed using the cDNA to measure the relative expression of Rag GTPase A (RAGA; Table 3), AKT, AMP-activated protein kinase alpha (AMPKα), insulin-like growth factor-I (IGF-I), eukaryotic initiation factor 4E binding protein 1 (eIF4EBP1), myosin heavy chain I (MHCI), myosin heavy chain IIA (MHCIIA), and myosin heavy chain IIX (MHCIIX). The RPS9 endogenous control gene was used to normalize relative abundance using the change in cycle threshold (ΔCT).

#### 2.2.5. Protein Extraction, Western Blotting, and Myosin Heavy Chain SDS-PAGE

The biopsy tissue fraction for total protein was processed to isolate total soluble protein using whole muscle extraction buffer (WMEB; 2% sodium dodecyl sulfate, 10 mM phosphate, pH 7.0). Five milliliters of WMEB were added to 1 g of longissimus sample and homogenized. The homogenate was centrifuged at 1500× *g* for 15 min and the middle aqueous layer was transferred to a microcentrifuge tube. The concentration of protein was determined using a commercially available assay (Pierce BCA; ThermoFisher Scientific, Waltham, MA, USA) and spectrophotometry (NanoDrop 1000; NanoDrop Products, Wilmington, DE, USA). All samples were diluted to the same concentration using WMEB. Western blots were prepared with modified Wangs Tracking Dye, and myosin heavy chain samples were prepared using Myosin Heavy Chain Tracking Dye and β-mercaptoethanol to denature the samples.

For Western blot analysis, 5 µg of each protein sample was loaded into precast Novex 4 to 12% Bis-Tris gels (Invitrogen, Carlsbad, CA, USA) and run for 35 min at 165V and 24 mA to separate proteins. The proteins were then transferred onto a nitrocellulose membrane for 7 min using an iBlot transfer device (Life Technologies). The membranes were incubated in a blocking solution of 1X Tris-buffered saline (TBS) and 5% non-fat dry milk (BIO RAD, Herculese, CA, USA) for 1 h at room temperature to block non-specific antibody binding. After blocking, a solution of TBS-Tween and the appropriate primary antibody [1:2000 α-RAGA (rabbit IgG; Abcam, Cambridge, MA, USA), 1:2500 α-AKT (rabbit IgG; Abcam), 1:1000 α-AMPKα (rabbit IgG; Cell Signaling Technology, Danvers, MA, USA), 1:1000 α-phosphorylated AMPKα (rabbit IgG; Cell Signaling Technology), 1:2500 α-Raptor (rabbit IgG; Abcam), or 1:1000 α-eIF4EBP1 (rabbit IgG; Abcam)] were added and allowed to incubate overnight at 4 °C on a rocker plate. The membranes were then rinsed three times with TBS-Tween for 10 min each at room temperature. Secondary antibody, comprised of 1:2000 Alexa-Fluor 633 (goat α-rabbit IgG; Invitrogen) in TBS-Tween, was applied and membranes were incubated in the dark for 1 h at room temperature. The membranes were rinsed three times with TBS-Tween for 10 min each in the dark at room temperature. Finally, the membranes were allowed to dry in the dark and imaged using Imager Scanner II and ImageQuant TL software. The densitometry measurements were taken on the bands corresponding to RAGA, AKT, AMPKα, pAMPKα, Raptor, and eIF4EBP1 based on a molecular weight standard reference which was run alongside the samples (Precision Plus Protein, All Blue Standards; BIO-RAD, Hercules, CA, USA).

To analyze the expression of myosin heavy chain protein, 4 to 6% polyacrylamide gels were cast and allowed to polymerize overnight. Myosin heavy chain protein samples were incubated at 95 °C for 5 min, loaded into the gel, and run for 30 h at 100 V. Gels were removed and placed in Coomassie Fluor Orange (Life Technologies) and incubated for 30 min at room temperature in the dark. The gels were then rinsed in 7.5% acetic acid and incubated in NanoPure water for 5 min in the dark at room temperature. Finally, gels were imaged using the aforementioned software and densitometry measurements were taken on the bands representing MHC-I and MHC-II.

#### 2.2.6. Immunohistochemical Analysis

Embedded tissue samples were thawed at −20 °C for 24 h for immunohistochemical staining to determine muscle fiber distribution, area, and satellite cell quantity. Samples were removed from the cork, sliced into 10-µm thick cross-sections at −20 °C with a cryostat (Leica CM1950; Leica Biosystems, Buffalo Grove, IL, USA), and placed on positively charged glass slides (4 slides per sample, 5 cross-sections per slide; Superfrost Plus, VWR International). Cross-sectional samples were fixed with 4% paraformaldehyde in phosphate-buffered saline (PBS; Thermo Fisher Scientific) for 10 min at room temperature, then briefly rinsed twice in PBS followed by one 10 min rinse in PBS at room temperature. A blocking solution consisting of 2% bovine serum albumin (MO Biomedical, Solon, OH), 5% horse serum (Invitrogen), and 0.2% Triton-X 100 (ThermoFisher Scientific) in PBS was then applied to the slides and allowed to incubate for 30 min at room temperature to prevent non-specific antibody binding. Slides were then incubated with the following primary antibodies in blocking solution for 1 h at room temperature: slide (1) 1:100 α-dystrophin (rabbit IgG; Thermo Scientific), 1:100 supernatant anti-MHCI IgG2b (BA-D5; Developmental Studies Hybridoma Bank, University of Iowa, Iowa City, IA, USA), and supernatant anti-MHC all but type IIX IgG1 (BF-35, Developmental Studies Hybridoma Bank); and slide (2) 1:10 supernatant anit-PAX7 (mouse α-chicken; Developmental Studies Hybridoma Bank), and 1:100 Myf-5 (rabbit IgG; Santa Cruz Biotechnology, Dallas, TX, USA). The slides were rinsed three times in PBS for 5 min at room temperature. The following secondary antibodies in blocking solution were then applied to the cross-sections for 30 min at room temperature in the dark: slide (1) 1:1000 Alexa-Fluor 488 (goat α-rabbit IgG; Invitrogen), 1:1000 Alexa-Fluor 546 (goat α-mouse IgG1; Invitrogen), 1:1000 Alexa-Fluor 633 (goat α-mouse IgG2b; Invitrogen), and slide (2) 1:1000 Alexa-Fluor 488 (goat α-rabbit IgG; Invitrogen), and 1:1000 Alexa-Fluor 546 (goat α-mouse IgG1; Invitrogen). Following the incubation period, the slides were again rinsed three times in PBS for 5 min at room temperature. The slides used for muscle fiber type and area were then cover-slipped using thin glass coverslips (VWR International) and ProLong^®^ Gold with 4’6-diamidino-2-phenylindole (DAPI) mounting media (Life Technologies) and allowed to cure in the dark for 36 h at room temperature. Slides used for satellite cell abundance were incubated with 1 µg/mL DAPI (ThermoFisher Scientific) for 1 min and rinsed twice briefly in PBS. The slides were then cover-slipped with AquaMount mounting media (Lerner Laboratories, Pittsburgh, PA) and thin glass coverslips (VWR International) and cured for 24 h at 4 °C in the dark. All slides were imaged within 48 h of curing. The slides were imaged at 20X magnification with an inverted fluorescence microscope (Nikon Eclipse, Ti-E; Nikon Instruments Inc., Melville, NY, USA) using a UV light source (Nikon Intensilight Inc., Melville, NY, USA; C-HGFIE), and images were captured using a CoolSnap ES2 monochrome camera. Images were artificially colored and analyzed with NIS Elements Imaging software.

#### 2.2.7. Statistical Analysis

All statistical analyses were performed using SAS (SAS version 9.3; SAS Institute, Inc., Cary, NC, USA). The MIXED procedure was used to analyze gene and protein expression, and the individual animal was used as the experimental unit. Treatment was included as a fixed effect and weight block was considered a random effect. The GLIMMIX procedure was used to analyze immunohistochemical data using the same fixed and random effects. Day was used as a repeated measure and the Kenward–Roger adjustment was used to correct the degrees of freedom for all data. Results are reported as least squares means and were separated using the PDIFF option of SAS. Means were determined to be significantly different when *p* ≤ 0.05. For all analyses, an alpha level ≤ 0.05 was considered significant, and tendencies were declared for values between 0.05 and 0.10.

## 3. Results and Discussion

### 3.1. Experiment 1

There were no significant differences in any measure of live performance when cattle were provided supplemental methionine (Table 4). There was a tendency for methionine supplementation to increase the average daily gain (ADG) during the final 28 d (*p* = 0.09), however, this trend was not observed for the duration of the feeding period, and ADG was numerically lower in all treatment groups when compared to controls. Previously reported results of methionine supplementation in finishing beef cattle are conflicted. Hussein and Berger [12] failed to elicit a significant response in performance when cattle were supplemented with increasing levels of a ruminally protected lysine-methionine blend. However, [5] reported a significant improvement in ADG and an additional 10 kg of final live weight when steers were supplied with 4 g/(head d) of supplemental methionine. Based on currently reported results and inconsistency in previously conducted studies, the lack of performance response could be due to maximized methionine utilization in the energy-rich diet provided or may be because the experimental diets were not limiting in methionine. In a study evaluating the effect of post-ruminal amino acid supply on intake of lactating dairy cows, [13] found that methionine abomasally infused at 140% of its calculated intestinally absorbable requirement significantly reduced DMI and had negative impacts on measures of performance. Conversely, the observed response in the current study could be due to lysine and methionine levels being lower than required for a measurable impact, as levels fed in the present study were lower than reported in some previous studies [1,3].

Hot carcass weight was unaffected by the inclusion of supplemental methionine (*p* = 0.61), which agrees with the similar final BW observed across all treatments (Table 5). Although carcass weights were similar, there was a linear increase in LM area for increasing levels of methionine (*p* = 0.04). This equated to a 9% increase in LM area for steers supplemented with 12 g/(head d) of ruminally protected methionine when compared to controls. Additionally, methionine supplementation tended to reduce calculated yield grade (*p* = 0.08), which was likely a function of the improved LM area. All other carcass characteristics were similar among treatments. Previous work at the same institution showed increased HCW when cattle were supplemented with 12 g/(head d) of encapsulated lysine and 4 g/(head ·d) of encapsulated methionine, and similar improvements when supplemented with methionine alone [5]. Though some studies have reported impacts on carcass characteristics when cattle are fed elevated bypass methionine, results are conflicting, and others have reported similar carcass characteristics in supplemented and unsupplemented animals [12].

Though days of postmortem aging improved tenderness as measured by WBSF, the addition of supplemental methionine showed little impact on this metric (Table 6). No differences in shear force were observed between treatments in steaks aged 7, 14, or 21 d (*p* ≥ 0.27). Following 28 d of postmortem aging, steaks from supplemented cattle tended to have decreased estimation of tenderness as measured by WBSF with increasing levels of methionine (*p* = 0.09). This response agrees with previous work presented by [14] where similar reductions in tenderness were observed in the LM of lambs supplemented with methionine. Furthermore, evidence shows when cattle are supplemented with both ruminally protected lysine and methionine, tenderness decreases [5]. Still, [15] reported that steaks with a shear force ≤ 4.3 kg were considered acceptable to consumers 86% of the time, and [16] reported consumer acceptance 100% of the time when values were less than 3.9 kg of shear force. Although a linear increase in shear force was observed after 28-d of postmortem aging in the current study, all steaks fell well below the thresholds established in previous studies.

### 3.2. Experiment 2

No significant treatment × day interactions were observed for gene expression (*p* ≥ 0.10), so the following section will focus on the main effects of methionine supplementation and day (Table 7). There was a significant day effect for all genes of interest (*p* ≤ 0.02). The abundance of eIF4EBP1 mRNA decreased after d 0. The abundance of AKT, IGF-I, MHC-I, MHC-IIA, and MHC-IIX mRNA were greatest on d 42 and 56, and the RAGA mRNA abundance was greatest on d 56 compared to d 0. The observed day effect was expected due to the normally observed accretion of protein as animals approach physiological maturity. The increase in skeletal muscle mass as animals progress along the growth curve is due to increased mTOR signaling resulting in the augmented transcription and translation [17]. Though no significant treatment differences were observed, there was a tendency for reduced mRNA expression of AKT (*p* = 0.08) and increased MHC-I expression (*p* = 0.10) in the LM of methionine supplemented steers. Decreased expression of AKT could lead to a downregulation of mTORC1 which could inhibit downstream signaling for protein synthesis [18]. However, an increase in MHC-I gene expression would suggest a shift in muscle fiber types.

There were no significant treatment × day interactions (*p* ≥ 0.14) for protein expression as measured by Western blot (Table 8). Similar to gene expression results, there was a significant day effect for all proteins evaluated (*p* ≤ 0.04). Protein abundance of AKT was greatest on d 28, though abundance measured at all other time points was significantly lower than measured at d 0 (*p* ≤ 0.05). Activated protein kinase alpha (AMPKα) protein abundance was greatest at d 0 and 56 and were significantly depressed (*p* < 0.05) at d 28 and 42. Protein abundance of eIF4EBP1 was greatest on d 14 and 28, and pAMPKα abundance showed its greatest expression at d 42. Both Raptor and RAGA expression was greatest on d 42 and saw a general trend upward over time. Methionine supplementation did not significantly affect protein expression as measured by Western blot but tended to reduce the expression of Raptor when compared to 0MET (*p* = 0.08). In neonatal pigs, feeding a diet with balanced levels of amino acids led to increased levels of RAGA-mTOR interaction [19]. Level of phosphorylation of AKT, eIF4EBP1, RAGA, and Raptor were not measured; however, the functionality of these proteins due to methionine supplementation would be elucidated thusly. Activation of mTOR signaling pathway proteins by phosphorylation is necessary for subsequent effects on growth [20,21]. In bovine mammary cells, the addition of lysine and methionine resulted in increased levels of phosphorylated mTOR [6]. The abundance of both MHC-I and total MHC-II proteins in LM biopsy samples were greatest on d 56 (*p* < 0.05), but no treatment effect was observed.

The distribution of skeletal muscle fiber types is presented in Figure 1. A significant treatment × day interaction (*p* = 0.03) was observed for MHC-I fibers, as this fiber type was reduced in 0MET steers and maintained its abundance over time in cattle supplemented with encapsulated methionine. This reflects the tendency for increased MHC-I gene expression in 8MET steers. This response in cattle supplemented with both methionine and lysine is similar to results found by [5], where the combination of the two supplemented amino acids caused an increase in cross-sectional area of the muscle fibers but no change in the proportion of MHC-I fibers when compared to cattle supplemented with lysine only. No differences due to treatment, day, or the interaction thereof were detected in MHC-IIA fibers (*p* > 0.16). A significant day effect was observed for MHC-IIX fibers (*p* = 0.03), with the proportion of MHC-IIX fibers increasing from d 0 to 56, which agrees with [22]. A significant day effect (*p* < 0.01) was also observed for the cross-sectional area of all three fiber types (Table 9 and Figure 2). The cross-sectional area was significantly greater in all fiber types at d 56 when compared to the area measured at d 0 (*p* < 0.05). This data mimics the findings of [22], who concluded that fiber cross-sectional area increases as animals age.

No treatment × day interactions (*p* ≥ 0.16) were observed for nuclei and satellite cell density (Table 10). A tendency for a day effect on nuclei density was observed (*p* = 0.10). In the methionine supplemented group, an increase in myonuclei was observed (*p* = 0.05), meaning increased satellite cell incorporation into the muscle fiber [11,23]. This difference was not observed in previous work [5]. A significant day effect (*p* < 0.01) was observed for myonuclei, Myf5 positive satellite cells, PAX7 positive satellite cells, and dual expressing satellite cells. Over time, the density of myonuclei increased, with a 1.5-fold increase at d 56 when compared to d 0 (Figure 3). This was expected, as cattle rely on the incorporation of nuclei into the myofiber to sustain muscle growth [11]. The density of both Myf5 positive and dual expressing cells decreased from d 0 to d 56, while the density of PAX7 positive cells increased during this time. Supplementing steers with 8 g/(head d) of methionine significantly increased myonuclei density (*p* = 0.05). Methionine also tended to reduce the density of Myf5 expressing cells (*p* = 0.10) while decreasing the density of PAX7 positive satellite cells (*p* = 0.09). The increased density of myofibrillar nuclei in methionine supplemented cattle could explain the improved LM area that was observed in Exp. 1, as this incorporation into the cell is necessary for muscle hypertrophy to follow. It could be proposed that longissimus biopsy samples for Exp. 2 were collected before the increase in skeletal muscle fiber cross-sectional area could be observed, as Exp. 1 animals were supplemented for 111 or 139 d. Myf5 is considered to be involved in the regulation of the proliferation of satellite cells [24]. The observed tendency for increased density of PAX7 expressing satellite cells in the 8MET treatment suggested methionine supplementation increased proliferation and subsequent differentiation of satellite cells [24,25,26].

This theory of increased satellite cell fusion into the muscle fiber was supported by the increased density of myonuclei in methionine supplemented cattle. The decreased expression of dual positive satellite cells on d 42 and d 56 for both treatments suggested an increase in the fusion of satellite cells into the muscle fiber. The increase PAX7 positive satellite cells seen at d 42 and 56 could result in increased expression of dual positive cells, as PAX7 positive satellite cells can produce dual positive daughter satellite cells [27]. The combination of increased satellite cell fusion and proliferation are indicators of potential muscle hypertrophy, as satellite cells tend to fuse into the muscle fiber proceeding and concurrently with muscle protein synthesis [28]. This is supported by the increase in density of nuclei not associated with satellite cells that have appeared to fuse into the existing muscle fiber to support post-natal skeletal muscle hypertrophy.

## 4. Conclusions

Overall, supplementation of methionine did not impact live performance, but increased longissimus muscle area, leading to a tendency of improved yield grades. Methionine supplementation also increased shear force values at 28 d of aging compared to control, however, all values fell well below shear force thresholds established for consumer acceptance of beef loin steaks. There were tendencies for change in gene and protein expression for methionine supplementation and coupled with skeletal muscle fiber type immunohistochemical data suggest methionine supplementation may prevent the transition of fibers from MHC-I. Methionine supplementation did alter satellite cell activity, which indicated that treatment with methionine harbors the potential for increased satellite cell incorporation into the muscle fiber. Further research evaluating phosphorylation of mTOR signaling proteins and immunohistochemical changes in satellite cell activity through the entire feeding period could provide increased insight into the effects of methionine supplementation on the growth of skeletal muscle via mTOR pathway signaling.

## Figures and Tables

**Figure 1 animals-11-01627-f001:**
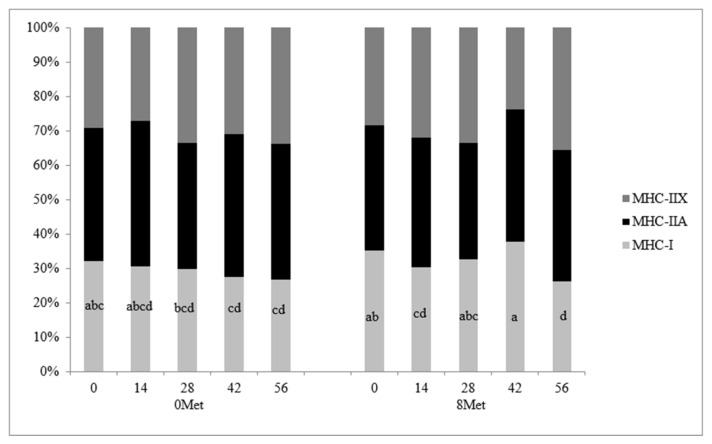
The distribution of skeletal muscle fiber types classified by myosin heavy chain (MHC) isoform in bovine longissimus tissue biopsy samples collected on d 0, 14, 28, 42, and 56 of the feeding trial (*n* = 20). Treatments were 0MET (0 g of encapsulated methionine and 4 g of encapsulated lysine per head per d) and 8MET (8 g of encapsulated methionine and 4 g of encapsulated lysine per head per d). Cross-sections of skeletal muscle samples were stained by immunohistochemistry for the presence of MHC isoforms. Differing superscripts denote means within MHC isoform I differ for treatment by day interaction (*p* = 0.03).

**Figure 2 animals-11-01627-f002:**
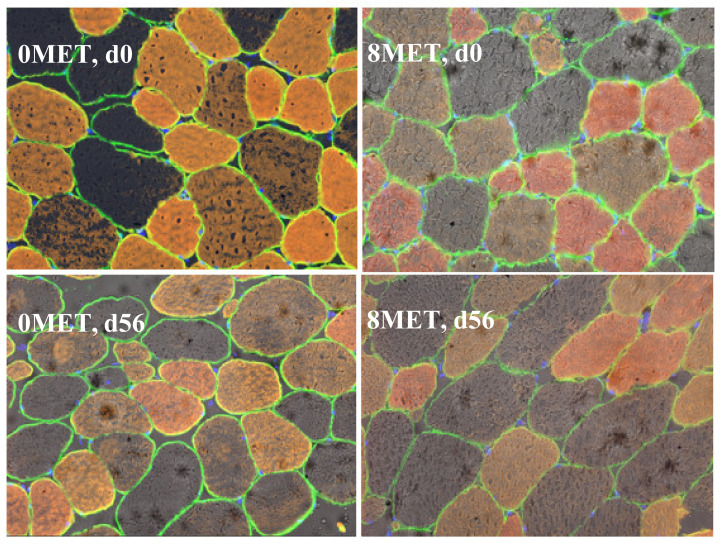
Immunohistochemical detection of muscle fiber type distribution and cross-sectional area in *longissimus* tissue. Red = type I fibers, yellow = type IIA fibers, gray/no stain = type IIX fibers, green = sarcolemma, blue = nuclei. 0MET = 0 g encapsulated methionine and 4 g encapsulated lysine per hd/d, 8MET = 8 g encapsulated methionine and 4 g encapsulated lysine per hd/d.

**Figure 3 animals-11-01627-f003:**
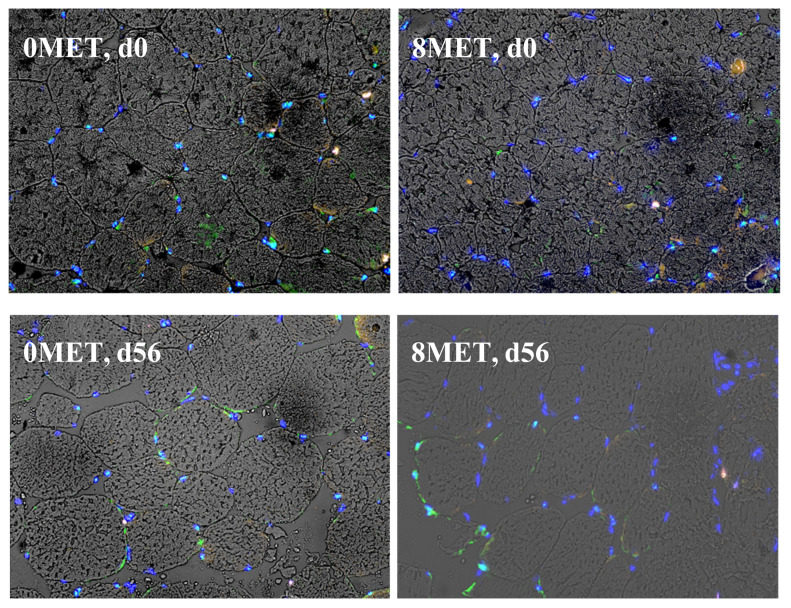
Immunohistochemical detection of satellite cells in longissimus tissue. Yellow = PAX7, green = Myf5, blue = nuclei. 0MET = 0 g encapsulated methionine and 4 g encapsulated lysine per hd/d, 8MET = 8 g encapsulated methionine and 4 g encapsulated lysine per hd/d.

**Table 1 animals-11-01627-t001:** Ingredient composition of experimental diet (%, dry matter [DM] basis).

Ingredient	DM %
Corn, steam flaked	75.24
Cottonseed hulls	5.95
Cottonseed meal	4.71
Alfalfa hay, chopped	3.93
Molasses	3.84
TTU supplement ^1^	1.97
Fat	1.96
Limestone	1.26
Chromium propionate supplement ^2^	0.25

^1^ Supplement composition (DM basis): 66.476% cottonseed meal; 15.000% salt; 10.000% potassium chloride; 4.167% ammonium sulfate; 0.986% zinc sulfate; 0.750% Rumensin (176.4 mg/kg; Elanco Animal Health, Indianapolis, IN, USA); 0.648% dicalcium phosphate; 0.563% Tylan (88.2 mg/kg; Elanco Animal Health); 0.500% Endox (Kemin Industries, Des Moines, IA, USA); 0.333% manganese oxide; 0.196% copper sulfate; 0.158% vitamin E (500 IU/g; DSM Nutritional Products, Inc., Parsippany, NJ, USA); 0.125% selenium premix (0.2% Se); 0.083% iron sulfate; 0.010% vitamin A (1,000,000 IU/g; DSM Nutritional Products, Inc., Parsippany); 0.003% ethylenediamine dihydroiodide; and 0.002% cobalt carbonate; ^2^ KemTraceTM (Kemin Industries, Des Moines, IA, USA). ^2^ KemTraceTM (Kemin Industries, Des Moines, IA, USA).

**Table 2 animals-11-01627-t002:** Amino acid composition of experimental ration ^1^.

Amino Acid.	%, DM
Alanine.	0.64
Arginine.	0.50
Aspartic Acid.	0.74
Cysteine.	0.18
Glutamic Acid.	1.68
Glycine.	0.37
Histidine.	0.24
Hydroxylysine.	0.05
Hydroxyproline.	0.07
Isoleucine.	0.36
Lanthionine.	0.00
Leucine.	1.04
Lysine.	0.37
Methionine.	0.16
Ornithine.	0.01
Phenylalanine.	0.46
Proline.	0.69
Serine.	0.40
Taurine.	0.16
Threonine.	0.32
Tryptophan.	0.13
Tyrosine.	0.32
Valine.	0.45
**Total.**	**9.34**
**Crude Protein ^2^.**	**12.47**

^1^ Measured as grams per 100 g sample; ^2^ Crude Protein = Percent Nitrogen × 6.25.

**Table 3 animals-11-01627-t003:** Sequence for bovine PCR primers and TaqMan probes for myogenic genes ^1^.

Item	Sequence (5′ to 3′)
AKT (accession no. AF207874)	
Forward	TTGCCCATAACTAAGCCTACATCTC
Reverse	CATGCGTGCCATTTGTTGAC
TaqMan probe	6FAM-TGCCCCAGCAACAC-TAMRA
AMPK-α (accession no. NM_001109802)	
Forward	ACCATTCTTGGTTGCTGAAACTC
Reverse	CACCTTGGTGTTTGGATTTCTG
TaqMan probe	6FAM-CAGGGCGCGCCATACCCTTG-TAMRA
eIF4EBP1 (accession no. NM_001077893)	
Forward	GGCGGCACGCTCTTCA
Reverse	AGGAACTTCCGGTCATAGATGATC
TaqMan probe	6FAM-ACCCCTGGAGGTACC-TAMRA
IGF-I (accession no. X15726)	
Forward	TGTGATTTCTTGAAGCAGGTGAA
Reverse	AGCACAGGGCCAGATAGAAGAG
TaqMan probe	6FAM-TGCCCATCACATCCTCCTCGCA-TAMRA
MHC I (accession no. AB059400)	
Forward	CCCACTTCTCCCTGATCCACTAC
Reverse	TTGAGCGGGTCTTTGTTTTTCT
TaqMan probe	6FAM-CCGGCACGGTGGACTACAACATCATAG-TAMRA
MHC IIA (accession no. AB059398)	
Forward	GCAATGTGGAAACGATCTCTAAAGC
Reverse	GCTGCTGCTCCTCCTCCTG
TaqMan probe	6FAM-TCTGGAGGACCAAGTGAACGAGCTGA-TAMRA
MHC IIX (accession no. AB059399)	
Forward	GGCCCACTTCTCCCTCATTC
Reverse	CCGACCACCGTCTCATTCA
TaqMan probe	6FAM-CGGGCACTGTGGACTACAACATTACT-TAMRA
RPS9 (accession no. DT860044)	
Forward	GAGCTGGGTTTGTCGCAAAA
Reverse	GGTCGAGGCGGGACTTCT
TaqMan probe	6FAM-ATGTGACCCCGCGGAGACCCTTC-TAMRA
RRAGA (accession no. NM_001035499)	
Forward	GCACTCCCACGTCCGATT
Reverse	CGCCACAGTCCCACAGATT
TaqMan probe	6FAM-CTGGGCAACCTAGTGC-TAMRA

^1^ AKT = AKT/protein kinase B, AMPKα = AMP-activated protein kinase alpha, eIF4EBP1 = eukaryotic initiation factor 4E binding protein 1, IGF-I = insulin-like growth factor I, MHC-I = myosin heavy chain-I, MHC-IIA = myosin heavy chain-IIA, MHC-IIX = myosin heavy chain-IIX, and RAGA = Rag GTPase A.

**Table 4 animals-11-01627-t004:** Effects of encapsulated methionine (MET) supplementation on feedlot performance.

Item ^2^	Treatment ^1^				SEM	Contrasts (*p*-Value)	
	0 MET	4 MET	8 MET	12 MET		Linear	0 v MET
Initial BW, kg	341	341	341	342	13.5	0.33	0.74
Pre-RH BW, kg	580	575	574	576	12.6	0.69	0.55
Final BW, kg	622	611	612	616	13.2	0.64	0.36
Pre-RH period ^3^							
ADG, kg	2.25	2.21	2.19	2.21	0.060	0.57	0.46
DMI, kg	10.34	9.93	10.09	10.15	0.260	0.66	0.25
G:F	0.218	0.223	0.217	0.218	0.0045	0.64	0.85
RH Period ^4^							
ADG, kg	1.54	1.30	1.38	1.43	0.090	0.52	0.09
DMI, kg	10.91	10.42	10.37	11.00	0.444	0.91	0.51
G:F	0.141	0.129	0.133	0.131	0.0092	0.41	0.22
Total							
ADG, kg	2.10	2.02	2.02	2.04	0.055	0.51	0.27
DMI, kg	10.46	10.03	10.15	10.33	0.286	0.78	0.28
G:F	0.201	0.202	0.199	0.198	0.0049	0.57	0.89

^1^ Treatments were 0, 4, 8, or 12 g/(head d) of methionine supplied to the small intestine; ^2^ A 3% shrink was applied to all bodyweight measurements; ^3^ Represents d 0 to 99 for first harvest group (*n* = 62) or d 0 to 111 for second harvest group (*n* = 61); ^4^ Represents d 100 to 127 for first harvest group (*n* = 62) or d 112 to 139 for second harvest group (*n* = 61).

**Table 5 animals-11-01627-t005:** Effects of encapsulated methionine (MET) supplementation on carcass characteristics.

Item	Treatment ^1^				SEM	Contrasts (*p*-Value)	
	0 MET	4 MET	8 MET	12 MET		Linear	0 v MET
HCW, kg	396	391	393	394	8.8	0.92	0.61
Dressing Percent ^2^	63.7	63.9	64.2	64.0	0.00	0.44	0.41
LM Area, cm ^2^	90.1	93.9	92.6	98.1	2.64	0.04	0.08
Marbling Score ^3^	432	402	420	420	14.9	0.79	0.31
Fat Thickness, cm	1.32	1.19	1.26	1.32	0.065	0.81	0.34
KPH, %	2.01	2.01	2.10	2.10	0.055	0.18	0.36
Yield Grade ^4^	3.1	2.7	2.9	2.7	0.17	0.15	0.08

^1^ Treatments were 0, 4, 8, or 12 g/(head d) of methionine supplied to the small intestine; ^2^ Dressing Percent = (HCW/shrunk final BW) × 100; ^3^ As determined by camera data: 300 = Slight00, 400 = Small00; ^4^ Calculated using USDA yield grade equation YG = 2.50 + (2.5 × adjusted fat thickness in inches) + (0.2 × percent KPH) + (0.0038 × HCW) − (0.32 × LM Area in inches).

**Table 6 animals-11-01627-t006:** Effects of encapsulated methionine (MET) supplementation on Warner–Bratzler Shear Force (WBSF) value.

WBSF, kg	Treatment ^1^				SEM	Contrasts (*p*-Value)	
	0 MET	4 MET	8 MET	12 MET		Linear	0 v MET
7 d aged	3.43	3.55	3.70	3.72	0.216	0.21	0.27
14 d aged	3.15	3.54	3.23	3.52	0.219	0.44	0.29
21 d aged	3.10	3.36	3.13	3.52	0.171	0.22	0.27
28 d aged	2.87	2.79	2.98	3.12	0.152	0.09	0.46

^1^ Treatments were 0, 4, 8, or 12 g/(head d) of methionine supplied to the small intestine.

**Table 7 animals-11-01627-t007:** Effect of encapsulated amino acids on relative mRNA abundances of AKT, AMPKα, eIF4EBP1, IGF-I, MHC-I, MHC-IIA, MHC-IIX, and RAGA genes in longissimus tissue ^1^.

Gene *^,3^	Treatment ^2^		SEM	Day					SEM	*p*-Value		
	0 MET	8 MET		0	14	28	42	56		TRT	Day	TRT × Day
AKT	0.89	0.78	0.067	0.73 ^c^	0.76 ^bc^	0.58 ^c^	1.12 ^a^	0.98 ^ab^	0.086	0.08	<0.01	0.16
AMPkα	1.25	1.14	0.113	0.72 ^b^	0.65 ^b^	0.53 ^b^	2.08 ^a^	2.00 ^a^	0.130	0.27	<0.01	0.10
eIF4EBP1	0.68	0.73	0.081	0.84 ^a^	0.67 ^ab^	0.70 ^ab^	0.59 ^b^	0.74 ^ab^	0.097	0.65	0.02	0.66
IGF-I	2.99	3.04	0.267	1.12 ^b^	0.83 ^b^	1.11 ^b^	6.80 ^a^	5.21 ^a^	0.337	0.76	<0.01	0.20
MHC-I	1.76	2.00	0.149	0.91 ^b^	0.93 ^b^	0.71 ^c^	3.71 ^a^	3.15 ^a^	0.181	0.10	<0.01	0.95
MHC-IIA	1.14	1.06	0.127	0.45 ^b^	0.51 ^b^	0.47 ^b^	2.01 ^a^	2.07 ^a^	0.126	0.59	<0.01	0.71
MHC-IIX	0.65	0.66	0.081	0.45 ^b^	0.44 ^b^	0.32 ^c^	1.03 ^a^	1.03 ^a^	0.080	0.92	<0.01	0.62
RAGA	1.66	1.56	0.114	1.12 ^c^	1.12 ^c^	1.11 ^c^	1.96 ^b^	2.74 ^a^	0.123	0.34	<0.01	0.61

^a–c^ Means within a row differ (*p* < 0.05); * AKT = AKT/protein kinase B, AMPKα = AMP-activated protein kinase alpha, eIF4EBP1 = eukaryotic initiation factor 4E binding protein 1, IGF-I = insulin-like growth factor I, MHC-I = myosin heavy chain-I, MHC-IIA = myosin heavy chain-IIA, MHC-IIX = myosin heavy chain-IIX, and RAGA = Rag GTPase A; ^1^ Reported in arbitrary units calibrated using a single sample; ^2^ Treatments were 0 or 8 g/(head d) of methionine supplied to the small intestine; ^3^ Relative abundance of the genes were normalized with the RPS9 endogenous control using the change in cycle threshold (ΔCT).

**Table 8 animals-11-01627-t008:** Effect of encapsulated amino acids on relative protein abundances of AKT, AMPKα, eIF4EBP1, pAMPKα, Raptor, and RAGA proteins in longissimus tissue ^1^.

Protein ^3^	Treatment ^2^		SEM	Day					SEM	*p*-Value		
	0 MET	8 MET		0	14	28	42	56		TRT	Day	TRT × Day
AKT	4140	3875	111.9	4342 ^ab^	3707 ^b^	5656 ^a^	3349 ^b^	2984 ^b^	121.3	0.34	0.04	0.86
AMPkα	4247	4134	64.4	4358 ^ab^	4296 ^bc^	4033 ^dc^	3710 ^d^	4555 ^a^	86.6	0.23	<0.01	0.63
eIF4EBP1	5927	5733	137.3	5559 ^b^	6578 ^a^	636 7 ^a^	5252 ^b^	5395 ^b^	166.7	0.25	<0.01	0.79
pAMPKα	6218	5695	290.1	7980 ^ab^	3897 ^b^	5624 ^ab^	8591 ^a^	3692 ^b^	338.2	0.42	0.03	0.49
MHC-I	39,698	40,557	1664.8	40,146 ^c^	27,411 ^e^	32,653 ^d^	46,376 ^b^	54,052 ^a^	1552.9	0.72	<0.01	0.14
MHC-II	32,504	32,423	1639.7	31,595 ^c^	22,840 ^e^	26,561 ^d^	37,894 ^b^	43,427 ^a^	1458.2	0.97	<0.01	0.95
Raptor	4332	4163	69.0	3137 ^d^	4664 ^c^	2579 ^e^	5692 ^a^	5167 ^b^	97.7	0.08	<0.01	0.14
RAGA	4491	4403	79.8	2944 ^d^	4107 ^c^	5147 ^a^	5248 ^a^	4788 ^b^	101.7	0.44	<0.01	0.77

^a–e^ Means within a row differ (*p* < 0.05); ^1^ Reported in relative light units per second; ^2^ Treatments were 0 or 8 g/(head d) of methionine supplied to the small intestine; ^3^ AKT = AKT/protein kinase B, AMPKα = AMP-activated protein kinase alpha, eIF4EBP1 = eukaryotic initiation factor 4E binding protein 1, pAMPkα = phosphorylated AMPkα, MHC-I = myosin heavy chain type I, MHC-II = myosin heavy chain type IIA and IIX, Raptor = regulatory protein associated with mammalian target of rapamycin complex 1, and RAGA = Rag GTPase A.

**Table 9 animals-11-01627-t009:** Effect of encapsulated amino acids on muscle fiber cross-sectional area in longissimus tissue.

Area *	Treatment ^1^		SEM	Day					SEM	*p*-Value		
	0 MET	8 MET		0	14	28	42	56		TRT	Day	TRT × Day
MHC-I, µm^2^	4212	4137	354.8	3667 ^b^	4245 ^b^	3613 ^b^	4264 ^ab^	5083 ^a^	434.4	0.83	<0.01	0.36
MHC-IIA, µm^2^	6082	6283	439.5	5157 ^c^	5983 ^bc^	5020 ^bc^	7100 ^ab^	7654 ^a^	700.7	0.65	<0.01	0.57
MHC-IIX, µm^2^	7790	7397	517.7	6654 ^bc^	7823 ^ab^	6278 ^c^	8925 ^a^	8289 ^a^	645.0	0.46	<0.01	0.51

^a–c^ Means within a row differ (*p* < 0.05); * MHC-I = myosin heavy chain-I, MHC-IIA = myosin heavy chain-IIA, and MHC-IIX = myosin heavy chain-IIX.; ^1^ Treatments were 0 or 8 g/(head d) of methionine supplied to the small intestine.

**Table 10 animals-11-01627-t010:** Effect of encapsulated amino acids on muscle nuclei and satellite cell density in longissimus tissue.

Type per mm^2^ *	Treatment ^1^		SEM	Day					SEM	*p*-Value		
	0 MET	8 MET		0	14	28	42	56		TRT	Day	TRT × Day
Nuclei	482.6	493.3	17.82	459.3	474.6	525.1	510.9	469.8	38.89	0.55	0.10	0.19
Myonuclei	372.9 ^y^	400.6 ^x^	13.90	305.1 ^d^	341.1 ^cd^	382.2 ^cd^	435.8 ^ab^	469.6 ^a^	43.55	0.05	<0.01	0.30
Myf5	94.9	77.6	10.14	136.0 ^a^	119.4 ^a^	128.6 ^a^	24.7 ^b^	22.5 ^b^	17.63	0.10	<0.01	0.16
PAX7	4.9	6.2	0.72	2.9 ^c^	2.7 ^c^	3.9 ^c^	11.1^a^	7.1 ^b^	1.57	0.09	<0.01	0.85
Myf5/PAX7	9.9	8.9	0.86	15.3 ^a^	11.4 ^b^	10.4 ^b^	5.4 ^c^	4.4 ^c^	1.94	0.27	<0.01	0.26

^a–d,x,y^ Means within a row differ (*p* < 0.05); * Nuclei = total nuclei, Myonuclei = nuclei associated with the muscle fiber, Myf5 = satellite cells expressing only Myf5, PAX7 = satellite cells expressing only PAX7, Myf5/PAX7= satellite cells expressing both Myf5 and PAX7; ^1^ Treatments were 0 or 8 g/(head d) of methionine supplied to the small intestine.

## Data Availability

Not applicable.

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
