# Peer review of "Effects of Encapsulated Methionine on Skeletal Muscle Growth and Development and Subsequent Feedlot Performance and Carcass Characteristics in Beef Steers"

_animals, 2021, doi:10.3390/ani11061627_

Round 1
Reviewer 1 Report
Manuscript ID: animals-1201778
Title: Effects of encapsulated methionine on skeletal muscle growth and development and subsequent feedlot performance and carcass characteristics in beef steers
Authors: Jessica Baggerman, Alex Thompson, Michael Jennings, Jerilyn Hergenreder, Whitney Rounds, Zachary Smith, Bradley J Johnson
The authors presented an interesting manuscript on the effect of encapsulated methionine on skeletal muscle development and parameters related to feeding efficiency and the characteristics of beef carcasses.
The research was performed with due diligence, the manuscript is written well, the materials and methods used are extensively presented, which will allow for their reliable use when trying to repeat or plan research with a similar methodology.
How did the authors choose the methionine doses used in the presented study and why they used doses lower than those indicated in other studies - which could significantly change the results of the obtained studies (Line 378-381)?
Minor issues:
No explanation of the abbreviations used: BW, HCW, DM, DMI, LM, KPH, ADG, others…?
Editorial corrections:
Duplicate spaces in lines: 67, 83, 97, 171, 173, 177, 378, 380, 412, 431, 457, 459, 462, 502, 503, 522
Table 6: lines other than those used for other tables
It would be more legible to put short descriptions of individual photos on themselves (e.g. 8MET, d0) – white color.
The authors sometimes use shortcuts written as 8MET, other times 8Met - it should be standardized.
Hardly legible, dark Figure 3.
Author Response
Reviewer 1:
- Comments and Suggestions for Authors
- Manuscript ID: animals-1201778
- Title: Effects of encapsulated methionine on skeletal muscle growth and development and subsequent feedlot performance and carcass characteristics in beef steers
- Authors: Jessica Baggerman, Alex Thompson, Michael Jennings, Jerilyn Hergenreder, Whitney Rounds, Zachary Smith, Bradley J Johnson
- The authors presented an interesting manuscript on the effect of encapsulated methionine on skeletal muscle development and parameters related to feeding efficiency and the characteristics of beef carcasses.
- The research was performed with due diligence, the manuscript is written well, the materials and methods used are extensively presented, which will allow for their reliable use when trying to repeat or plan research with a similar methodology.
- How did the authors choose the methionine doses used in the presented study and why they used doses lower than those indicated in other studies - which could significantly change the results of the obtained studies (Line 378-381)?
- Levels were selected as ratios to the 4g lysine/hd/d, see lines 122-123 in manuscript for clarification
- Minor issues:
- No explanation of the abbreviations used: BW, HCW, DM, DMI, LM, KPH, ADG, others…?
- Addressed in manuscript
- Editorial corrections:
- Duplicate spaces in lines: 67, 83, 97, 171, 173, 177, 378, 380, 412, 431, 457, 459, 462, 502, 503, 522
- Addressed in manuscript
- Table 6: lines other than those used for other tables
- Addressed in manuscript
- It would be more legible to put short descriptions of individual photos on themselves (e.g. 8MET, d0) – white color.
- Addressed in manuscript
- The authors sometimes use shortcuts written as 8MET, other times 8Met - it should be standardized.
- Addressed in manuscript
- Hardly legible, dark Figure 3.
- Addressed in manuscript
- Duplicate spaces in lines: 67, 83, 97, 171, 173, 177, 378, 380, 412, 431, 457, 459, 462, 502, 503, 522
- No explanation of the abbreviations used: BW, HCW, DM, DMI, LM, KPH, ADG, others…?
Reviewer 2 Report
Comments on the paper animals-1201778 submitted for publication by Baggerman and co-workers.
This paper aims to investigate the role of encapsulated methionine on feedlot growth performance and carcass characteristics, and to elucidate the role of methionine supplementation on the mTOR signaling pathway and skeletal muscle development in feedlot steers during the finishing phase.
The paper is of very high scientific quality. However, some minor modifications are needed before I recommend this paper for publication.
- The paper is not fully following the template of the journal. Please check the whole manuscript.
- The abstract is long, not very informative. The authors should summarise it further.
- In contrast to the abstract, the introduction is very brief. It needs much work to further present what we know and what we do not in this topic. The hypothesis is not very clear.
- Some data such as Table 1 and Table 2 are not needed in the main manuscript that is already long. Please, I recommend to move that data to the supplementary section.
- I suggest to the authors to change “0 MET” by “Control”.
- The Figure 1 is not very clear. The evolution should be maybe better to highlight using curves over time or a Table.
- The increase in LM area should be carefully checked among individuals. The p-value is only 0.04 and not at all supported by muscle fiber cross-sectional area results presented in Table 9. Individual-variability should be carefully checked in this paper.
Author Response
Reviewer 2:
- Comments on the paper animals-1201778 submitted for publication by Baggerman and co-workers.
- This paper aims to investigate the role of encapsulated methionine on feedlot growth performance and carcass characteristics, and to elucidate the role of methionine supplementation on the mTOR signaling pathway and skeletal muscle development in feedlot steers during the finishing phase.
- The paper is of very high scientific quality. However, some minor modifications are needed before I recommend this paper for publication.
- The paper is not fully following the template of the journal. Please check the whole manuscript.
- The abstract is long, not very informative. The authors should summarise it further.
- Addressed in manuscript
- In contrast to the abstract, the introduction is very brief. It needs much work to further present what we know and what we do not in this topic. The hypothesis is not very clear.
- Wording changed to clarify hypothesis
- Some data such as Table 1 and Table 2 are not needed in the main manuscript that is already long. Please, I recommend to move that data to the supplementary section.
- Due to this manuscript being submitted to the nutrition section of Animals, the authors request all dietary information to remain in the main manuscript.
- I suggest to the authors to change “0 MET” by “Control”.
- The abbreviation 0MET was selected to be clear when discussing the treatment groups with the 4 levels of methionine supplementation in Exp. 1, and the same abbreviations were continued when discussing Exp. 2 for consistency.
- The Figure 1 is not very clear. The evolution should be maybe better to highlight using curves over time or a Table.
- A stacked proportion graph was selected for clarity compared to curves since this data is presented as percentages. When looking at the figure, the superscripts are denoted for MHC-I, which was the only muscle fiber type that showed any change in distribution as a percentage.
- The increase in LM area should be carefully checked among individuals. The p-value is only 0.04 and not at all supported by muscle fiber cross-sectional area results presented in Table 9. Individual-variability should be carefully checked in this paper.
- The LM area in table 5 was measured at harvest of Exp. 1, which was either on d 111 or d 139 of treatment, while the data in Table 9 was from samples collected until d 56 of treatment. Table 9 coupled with Table 10 suggests that possible increases in muscle fiber cross-sectional area due to methionine supplementation might have been observed if further biopsy samples had been collected. Lines 512-514 extrapolate this point.